# Estimation of the covariance structure of heavy-tailed distributions

**Stanislav Minsker**
Department of Mathematics
University of Southern California
Los Angeles, CA 90007
minsker@usc.edu

**Xiaohan Wei**
Department of Electrical Engineering
University of Southern California
Los Angeles, CA 90007
xiaohanw@usc.edu

## Abstract

We propose and analyze a new estimator of the covariance matrix that admits strong theoretical guarantees under weak assumptions on the underlying distribution, such as existence of moments of only low order. While estimation of covariance matrices corresponding to sub-Gaussian distributions is well-understood, much less in known in the case of heavy-tailed data. As K. Balasubramanian and M. Yuan write [1], "data from real-world experiments oftentimes tend to be corrupted with outliers and/or exhibit heavy tails. In such cases, it is not clear that those covariance matrix estimators .. remain optimal" and "..what are the other possible strategies to deal with heavy tailed distributions warrant further studies." We make a step towards answering this question and prove tight deviation inequalities for the proposed estimator that depend only on the parameters controlling the "intrinsic dimension" associated to the covariance matrix (as opposed to the dimension of the ambient space); in particular, our results are applicable in the case of high-dimensional observations.

## 1   Introduction

Estimation of the covariance matrix is one of the fundamental problems in data analysis: many important statistical tools, such as Principal Component Analysis (PCA, Hotelling, 1933) and regression analysis, involve covariance estimation as a crucial step. For instance, PCA has immediate applications to nonlinear dimension reduction and manifold learning techniques (Allard et al., 2012), genetics (Novembre et al., 2008), computational biology (Alter et al., 2000), among many others.

However, assumptions underlying the theoretical analysis of most existing estimators, such as various modifications of the sample covariance matrix, are often restrictive and do not hold for real-world scenarios. Usually, such estimators rely on heuristic (and often bias-producing) data preprocessing, such as outlier removal. To eliminate such preprocessing step from the equation, one has to develop a class of new statistical estimators that admit strong performance guarantees, such as exponentially tight concentration around the unknown parameter of interest, under weak assumptions on the underlying distribution, such as existence of moments of only low order. In particular, such heavy-tailed distributions serve as a viable model for data corrupted with outliers – an almost inevitable scenario for applications.

We make a step towards solving this problem: using tools from the random matrix theory, we will develop a class of *robust* estimators that are numerically tractable and are supported by strong theoretical evidence under much weaker conditions than currently available analogues. The term "robustness" refers to the fact that our estimators admit provably good performance even when the underlying distribution is heavy-tailed.

## 1.1 Notation and organization of the paper

Given $A \in \mathbb{R}^{d_1 \times d_2}$, let $A^T \in \mathbb{R}^{d_2 \times d_1}$ be transpose of $A$. If $A$ is symmetric, we will write $\lambda_{\max}(A)$ and $\lambda_{\min}(A)$ for the largest and smallest eigenvalues of $A$. Next, we will introduce the matrix norms used in the paper. Everywhere below, $\| \cdot \|$ stands for the operator norm $\|A\| := \sqrt{\lambda_{\max}(A^T A)}$. If $d_1 = d_2 = d$, we denote by $\mathrm{tr} A$ the trace of $A$. For $A \in \mathbb{R}^{d_1 \times d_2}$, the nuclear norm $\| \cdot \|_1$ is defined as $\|A\|_1 = \mathrm{tr}(\sqrt{A^T A})$, where $\sqrt{A^T A}$ is a nonnegative definite matrix such that $(\sqrt{A^T A})^2 = A^T A$. The Frobenius (or Hilbert-Schmidt) norm is $\|A\|_{\mathrm{F}} = \sqrt{\mathrm{tr}(A^T A)}$, and the associated inner product is $\langle A_1, A_2 \rangle = \mathrm{tr}(A_1^* A_2)$. For $z \in \mathbb{R}^d$, $\|z\|_2$ stands for the usual Euclidean norm of $z$. Let $A$, $B$ be two self-adjoint matrices. We will write $A \succeq B$ (or $A \succ B$) iff $A - B$ is nonnegative (or positive) definite. For $a, b \in \mathbb{R}$, we set $a \vee b := \max(a, b)$ and $a \wedge b := \min(a, b)$. We will also use the standard Big-O and little-o notation when necessary.

Finally, we give a definition of a matrix function. Let $f$ be a real-valued function defined on an interval $\mathbb{T} \subseteq \mathbb{R}$, and let $A \in \mathbb{R}^{d \times d}$ be a symmetric matrix with the eigenvalue decomposition $A = U \Lambda U^*$ such that $\lambda_j(A) \in \mathbb{T}$, $j = 1, \ldots, d$. We define $f(A)$ as $f(A) = U f(\Lambda) U^*$, where

$$f(\Lambda) = f\left(\begin{pmatrix} \lambda_1 & & \\ & \ddots & \\ & & \lambda_d \end{pmatrix}\right) := \begin{pmatrix} f(\lambda_1) & & \\ & \ddots & \\ & & f(\lambda_d) \end{pmatrix}.$$

Few comments about organization of the material in the rest of the paper: section 1.2 provides an overview of the related work. Section 2 contains the mains results of the paper. The proofs are outlined in section 4; longer technical arguments can be found in the supplementary material.

## 1.2 Problem formulation and overview of the existing work

Let $X \in \mathbb{R}^d$ be a random vector with mean $\mathbb{E} X = \mu_0$, covariance matrix $\Sigma_0 = \mathbb{E}\left[(X - \mu_0)(X - \mu_0)^T\right]$, and assume $\mathbb{E}\|X - \mu_0\|_2^4 < \infty$. Let $X_1, \ldots, X_m$ be i.i.d. copies of $X$. Our goal is to estimate the covariance matrix $\Sigma$ from $X_j$, $j \leq m$. This problem and its variations have previously received significant attention by the research community: excellent expository papers by Cai et al. (2016) and Fan et al. (2016) discuss the topic in detail. However, strong guarantees for the best known estimators hold (with few exceptions mentioned below) under the restrictive assumption that $X$ is either bounded with probability 1 or has sub-Gaussian distribution, meaning that there exists $\sigma > 0$ such that for any $v \in \mathbb{R}^d$ of unit Euclidean norm,

$$\mathrm{Pr}\left(|\langle v, X - \mu_0 \rangle| \geq t\right) \leq 2 e^{-\frac{t^2 \sigma^2}{2}}.$$

In the discussion accompanying the paper by Cai et al. (2016), Balasubramanian and Yuan (2016) write that "data from real-world experiments oftentimes tend to be corrupted with outliers and/or exhibit heavy tails. In such cases, it is not clear that those covariance matrix estimators described in this article remain optimal" and "..what are the other possible strategies to deal with heavy tailed distributions warrant further studies." This motivates our main goal: develop new estimators of the covariance matrix that (i) are computationally tractable and perform well when applied to heavy-tailed data and (ii) admit strong theoretical guarantees (such as exponentially tight concentration around the unknown covariance matrix) under weak assumptions on the underlying distribution. Note that, unlike the majority of existing literature, we do not impose *any further conditions* on the moments of $X$, or on the "shape" of its distribution, such as elliptical symmetry.

Robust estimators of covariance and scatter have been studied extensively during the past few decades. However, majority of rigorous theoretical results were obtained for the class of elliptically symmetric distributions which is a natural generalization of the Gaussian distribution; we mention just a small subsample among the thousands of published works. Notable examples include the Minimum Covariance Determinant estimator and the Minimum Volume Ellipsoid estimator which are discussed in (Hubert et al., 2008), as well Tyler's (Tyler, 1987) M-estimator of scatter. Works by Fan et al. (2016); Wegkamp et al. (2016); Han and Liu (2017) exploit the connection between Kendall's tau and Pearson's correlation coefficient (Fang et al., 1990) in the context of elliptical distributions to obtain robust estimators of correlation matrices. Interesting results for shrinkage-type estimators have been obtained by Ledoit and Wolf (2004); Ledoit et al. (2012). In a recent work, Chen et al. (2015) study Huber's $\varepsilon$-contamination model which assumes that the data is generated from the

distribution of the form $(1-\varepsilon)F + \varepsilon Q$, where $Q$ is an arbitrary distribution of "outliers" and $F$ is an elliptical distribution of "inliers", and propose novel estimator based on the notion of "matrix depth" which is related to Tukey's depth function (Tukey, 1975); a related class of problems has been studies by Diakonikolas et al. (2016). The main difference of the approach investigated in this paper is the ability to handle a much wider class of distributions that are not elliptically symmetric and only satisfy weak moment assumptions. Recent papers by Catoni (2016), Giulini (2015), Fan et al. (2016, 2017); Fan and Kim (2017) and Minsker (2016) are closest in spirit to this direction. For instance, Catoni (2016) constructs a robust estimator of the Gram matrix of a random vector $Z \in \mathbb{R}^d$ (as well as its covariance matrix) via estimating the quadratic form $\mathbb{E}\langle Z, u\rangle^2$ uniformly over all $\|u\|_2 = 1$. However, the bounds are obtained under conditions more stringent than those required by our framework, and resulting estimators are difficult to evaluate in applications even for data of moderate dimension. Fan et al. (2016) obtain bounds in norms other than the operator norm which the focus of the present paper (however, we plan to address optimality guarantees with respect to other norms in the future). Minsker (2016) and Fan et al. (2016) use adaptive truncation arguments to construct robust estimators of the covariance matrix. However, their results are only applicable to the situation when the data is centered (that is, $\mu_0 = 0$). In the robust estimation framework, rigorous extension of the arguments to the case of non-centered high-dimensional observations is non-trivial and requires new tools, especially if one wants to avoid statistically inefficient procedures such as sample splitting. We formulate and prove such extensions in this paper.

## 2 Main results

Definition of our estimator has its roots in the technique proposed by Catoni (2012). Let

$$\psi(x) = (|x| \wedge 1)\operatorname{sign}(x) \tag{1}$$

be the usual truncation function. As before, let $X_1, \ldots, X_m$ be i.i.d. copies of $X$, and assume that $\widehat{\mu}$ is a suitable estimator of the mean $\mu_0$ from these samples, to be specified later. We define $\widehat{\Sigma}$ as

$$\widehat{\Sigma} := \frac{1}{m\theta}\sum_{i=1}^{m}\psi\left(\theta(X_i - \widehat{\mu})(X_i - \widehat{\mu})^T\right), \tag{2}$$

where $\theta \simeq m^{-1/2}$ is small (the exact value will be given later). It easily follows from the definition of the matrix function that

$$\widehat{\Sigma} = \frac{1}{m\theta}\sum_{i=1}^{m}\frac{(X_i - \widehat{\mu})(X_i - \widehat{\mu})^T}{\|X_i - \widehat{\mu}\|_2^2}\psi\left(\theta\|X_i - \widehat{\mu}\|_2^2\right),$$

hence it is easily computable. Note that $\psi(x) = x$ in the neighborhood of 0; it implies that whenever all random variables $\theta\|X_i - \widehat{\mu}\|_2^2$, $1 \le i \le m$ are "small" (say, bounded above by 1) and $\hat{\mu}$ is the sample mean, $\widehat{\Sigma}$ is close to the usual sample covariance estimator. On the other hand, $\psi$ "truncates" $\|X_i - \widehat{\mu}\|_2^2$ on level $\simeq \sqrt{m}$, thus limiting the effect of outliers. Our results (formally stated below, see Theorem 2.1) imply that for an appropriate choice of $\theta = \theta(t, m, \sigma)$,

$$\left\|\widehat{\Sigma} - \Sigma_0\right\| \le C_0 \sigma_0 \sqrt{\frac{\beta}{m}}$$

with probability $\ge 1 - de^{-\beta}$ for some positive constant $C_0$, where

$$\sigma_0^2 := \left\|\mathbb{E}\|X - \mu_0\|_2^2 (X - \mu_0)(X - \mu_0)^T\right\|$$

is the "matrix variance".

### 2.1 Robust mean estimation

There are several ways to construct a suitable estimator of the mean $\mu_0$. We present the one obtained via the "median-of-means" approach. Let $x_1, \ldots, x_k \in \mathbb{R}^d$. Recall that the *geometric median* of $x_1, \ldots, x_k$ is defined as

$$\operatorname{med}(x_1, \ldots, x_k) := \operatorname*{argmin}_{z \in \mathbb{R}^d}\sum_{j=1}^{k}\|z - x_j\|_2.$$

Let $1 < \beta < \infty$ be the confidence parameter, and set $k = \left\lfloor 3.5\beta \right\rfloor + 1$; we will assume that $k \leq \frac{m}{2}$.
Divide the sample $X_1, \dots, X_m$ into $k$ disjoint groups $G_1, \dots, G_k$ of size $\left\lfloor \frac{m}{k} \right\rfloor$ each, and define

$$\hat{\mu}_j := \frac{1}{|G_j|} \sum_{i \in G_j} X_i, \ j = 1 \dots k,$$

$$\hat{\mu} := \mathrm{med}\left(\hat{\mu}_1, \dots, \hat{\mu}_k\right). \tag{3}$$

It then follows from Corollary 4.1 in (Minsker, 2015) that

$$\Pr\left( \|\hat{\mu} - \mu\|_2 \geq 11\sqrt{\frac{\mathrm{tr}(\Sigma_0)(\beta+1)}{m}} \right) \leq e^{-\beta}. \tag{4}$$

## 2.2 Robust covariance estimation

Let $\widehat{\Sigma}$ be the estimator defined in (2) with $\widehat{\mu}$ being the "median-of-means" estimator (3). Then $\widehat{\Sigma}$ admits the following performance guarantees:

**Lemma 2.1.** *Assume that $\sigma \geq \sigma_0$, and set $\theta = \frac{1}{\sigma}\sqrt{\frac{\beta}{m}}$. Moreover, let $\overline{d} := \sigma_0^2/\|\Sigma_0\|^2$, and suppose that $m \geq C\overline{d}\beta$, where $C > 0$ is an absolute constant. Then*

$$\left\| \widehat{\Sigma} - \Sigma_0 \right\| \leq 3\sigma\sqrt{\frac{\beta}{m}} \tag{5}$$

*with probability at least $1 - 5de^{-\beta}$.*

**Remark 2.1.** *The quantity $\overline{d}$ is a measure of "intrinsic dimension" akin to the "effective rank" $r = \frac{\mathrm{tr}(\Sigma_0)}{\|\Sigma_0\|}$; see Lemma 2.3 below for more details. Moreover, note that the claim of Lemma 2.1 holds for any $\sigma \geq \sigma_0$, rather than just for $\sigma = \sigma_0$; this "degree of freedom" allows construction of adaptive estimators, as it is shown below.*

The statement above suggests that one has to know the value of (or a tight upper bound on) the "matrix variance" $\sigma_0^2$ in order to obtain a good estimator $\widehat{\Sigma}$. More often than not, such information is unavailable. To make the estimator completely data-dependent, we will use Lepski's method (Lepski, 1992). To this end, assume that $\sigma_{\min}$, $\sigma_{\max}$ are "crude" preliminary bounds such that

$$\sigma_{\min} \leq \sigma_0 \leq \sigma_{\max}.$$

Usually, $\sigma_{\min}$ and $\sigma_{\max}$ do not need to be precise, and can potentially differ from $\sigma_0$ by several orders of magnitude. Set

$$\sigma_j := \sigma_{\min} 2^j \text{ and } \mathcal{J} = \{ j \in \mathbb{Z} : \ \sigma_{\min} \leq \sigma_j < 2\sigma_{\max} \}.$$

Note that the cardinality of $J$ satisfies $\mathrm{card}(\mathcal{J}) \leq 1 + \log_2(\sigma_{\max}/\sigma_{\min})$. For each $j \in \mathcal{J}$, define $\theta_j := \theta(j, \beta) = \frac{1}{\sigma_j}\sqrt{\frac{\beta}{m}}$. Define

$$\widehat{\Sigma}_{m,j} = \frac{1}{m\theta_j} \sum_{i=1}^{m} \psi\left(\theta_j(X_i - \widehat{\mu})(X_i - \widehat{\mu})^T\right).$$

Finally, set

$$j_* := \min\left\{ j \in \mathcal{J} : \forall k > j \text{ s.t. } k \in \mathcal{J}, \ \left\|\widehat{\Sigma}_{m,k} - \widehat{\Sigma}_{m,j}\right\| \leq 6\sigma_k\sqrt{\frac{\beta}{m}} \right\} \tag{6}$$

and $\widehat{\Sigma}_* := \widehat{\Sigma}_{m,j_*}$. Note that the estimator $\widehat{\Sigma}_*$ depends only on $X_1, \dots, X_m$, as well as $\sigma_{\min}$, $\sigma_{\max}$. Our main result is the following statement regarding the performance of the data-dependent estimator $\widehat{\Sigma}_*$:

**Theorem 2.1.** *Suppose $m \geq C\overline{d}\beta$, then, the following inequality holds with probability at least $1 - 5d\log_2\left(\frac{2\sigma_{max}}{\sigma_{min}}\right) e^{-\beta}$:*

$$\left\|\widehat{\Sigma}_* - \Sigma_0\right\| \leq 18\sigma_0\sqrt{\frac{\beta}{m}}.$$

An immediate corollary of Theorem 2.1 is the quantitative result for the performance of PCA based on the estimator $\widehat{\Sigma}_*$. Let $\mathrm{Proj}_k$ be the orthogonal projector on a subspace corresponding to the $k$ largest positive eigenvalues $\lambda_1, \ldots, \lambda_k$ of $\Sigma_0$ (here, we assume for simplicity that all the eigenvalues are distinct), and $\widehat{\mathrm{Proj}_k}$ – the orthogonal projector of the same rank as $\mathrm{Proj}_k$ corresponding to the $k$ largest eigenvalues of $\widehat{\Sigma}_*$. The following bound follows from the Davis-Kahan perturbation theorem (Davis and Kahan, 1970), more specifically, its version due to Zwald and Blanchard (2006, Theorem 3).

**Corollary 2.1.** *Let $\Delta_k = \lambda_k - \lambda_{k+1}$, and assume that $\Delta_k \geq 72\sigma_0\sqrt{\frac{\beta}{m}}$. Then*

$$\left\|\widehat{\mathrm{Proj}_k} - \mathrm{Proj}_k\right\| \leq \frac{36}{\Delta_k}\sigma_0\sqrt{\frac{\beta}{m}}$$

*with probability* $\geq 1 - 5d\log_2\left(\frac{2\sigma_{max}}{\sigma_{min}}\right)e^{-\beta}$.

It is worth comparing the bound of Lemma 2.1 and Theorem 2.1 above to results of the paper by Fan et al. (2016), which constructs a covariance estimator $\widehat{\Sigma}'_m$ under the assumption that the random vector $X$ is centered, and $\sup_{\mathbf{v}\in\mathbb{R}^d:\|\mathbf{v}\|_2\leq 1}\mathbb{E}\big[|\langle\mathbf{v},X\rangle|^4\big] = B < \infty$. More specifically, $\widehat{\Sigma}'_m$ satisfies the inequality

$$\mathbb{P}\left(\left\|\widehat{\Sigma}'_m - \Sigma_0\right\| \geq \sqrt{\frac{C_1\beta Bd}{m}}\right) \leq de^{-\beta}, \tag{7}$$

where $C_1 > 0$ is an absolute constant. The main difference between (7) and the bounds of Lemma 2.1 and Theorem 2.1 is that the latter are expressed in terms of $\sigma_0^2$, while the former is in terms of $B$. The following lemma demonstrates that our bounds are at least as good:

**Lemma 2.2.** *Suppose that $\mathbb{E}X = 0$ and $\sup_{\mathbf{v}\in\mathbb{R}^d:\|\mathbf{v}\|_2\leq 1}\mathbb{E}\big[|\langle\mathbf{v},X\rangle|^4\big] = B < \infty$. Then $Bd \geq \sigma_0^2$.*

It follows from the above lemma that $\bar{d} = \sigma_0^2/\|\Sigma_0\|^2 \lesssim d$. Hence, By Theorem 2.1, the error rate of estimator $\widehat{\Sigma}_*$ is bounded above by $\mathcal{O}(\sqrt{d/m})$ if $m \gtrsim d$. It has been shown (for example, Lounici, 2014) that the minimax lower bound of covariance estimation is of order $\Omega(\sqrt{d/m})$. Hence, the bounds of Fan et al. (2016) as well as our results imply correct order of the error. That being said, the "intrinsic dimension" $\bar{d}$ reflects the structure of the covariance matrix and can potentially be much smaller than $d$, as it is shown in the next section.

## 2.3 Bounds in terms of intrinsic dimension

In this section, we show that under a slightly stronger assumption on the fourth moment of the random vector $X$, the bound $\mathcal{O}(\sqrt{d/m})$ is suboptimal, while our estimator can achieve a much better rate in terms of the "intrinsic dimension" associated to the covariance matrix. This makes our estimator useful in applications involving high-dimensional covariance estimation, such as PCA. Assume the following uniform bound on the *kurtosis*:

$$\max_{k=1,2,\ldots,d}\frac{\sqrt{\mathbb{E}\left(X^{(k)} - \mu_0^{(k)}\right)^4}}{\mathbb{E}\left(X^{(k)} - \mu_0^{(k)}\right)^2} = R < \infty, \tag{8}$$

where $X^{(k)}$, $\mu_0^{(k)}$ denotes the $k$-th entry of $X$ and $\mu_0$ respectively. The intrinsic dimension of the covariance matrix $\Sigma_0$ can be measured by the *effective rank* defined as

$$\mathbf{r}(\Sigma_0) = \frac{\mathrm{tr}(\Sigma_0)}{\|\Sigma_0\|}.$$

Note that we always have $\mathbf{r}(\Sigma_0) \leq \mathrm{rank}(\Sigma_0) \leq d$, and it some situations $\mathbf{r}(\Sigma_0) \ll \mathrm{rank}(\Sigma_0)$, for instance if the covariance matrix is "approximately low-rank", meaning that it has many small eigenvalues. The constant $\sigma_0^2$ is closely related to the effective rank as is shown in the following lemma (the proof of which is included in the supplementary material):

**Lemma 2.3.** *Suppose that* (8) *holds. Then,*
$$\mathbf{r}(\Sigma_0)\|\Sigma_0\|^2 \leq \sigma_0^2 \leq R^2\mathbf{r}(\Sigma_0)\|\Sigma_0\|^2.$$

As a result, we have $\mathbf{r}(\Sigma_0) \leq \overline{d} \leq R^2\mathbf{r}(\Sigma_0)$. The following corollary immediately follows from Theorem 2.1 and Lemma 2.3:

**Corollary 2.2.** *Suppose that $m \geq C\beta\mathbf{r}(\Sigma_0)$ for an absolute constant $C > 0$ and that* (8) *holds. Then*
$$\left\|\widehat{\Sigma}_* - \Sigma_0\right\| \leq 18R\|\Sigma_0\|\sqrt{\frac{\mathbf{r}(\Sigma_0)\beta}{m}}$$
*with probability at least $1 - 5d\log_2\left(\frac{2\sigma_{max}}{\sigma_{min}}\right)e^{-\beta}$.*

## 3 Applications: low-rank covariance estimation

In many data sets encountered in modern applications (for instance, gene expression profiles (Saal et al., 2007)), dimension of the observations, hence the corresponding covariance matrix, is larger than the available sample size. However, it is often possible, and natural, to assume that the unknown matrix possesses special structure, such as low rank, thus reducing the "effective dimension" of the problem. The goal of this section is to present an estimator of the covariance matrix that is "adaptive" to the possible low-rank structure; such estimators are well-known and have been previously studied for the bounded and sub-Gaussian observations (Lounici, 2014). We extend these results to the case of heavy-tailed observations; in particular, we show that the estimator obtained via soft-thresholding applied to the eigenvalues of $\widehat{\Sigma}_*$ admits optimal guarantees in the Frobenius (as well as operator) norm.

Let $\widehat{\Sigma}_*$ be the estimator defined in the previous section, see equation (6), and set

$$\widehat{\Sigma}_*^\tau = \operatorname*{argmin}_{A \in \mathbb{R}^{d \times d}} \left[ \left\|A - \widehat{\Sigma}_*\right\|_{\mathrm{F}}^2 + \tau\|A\|_1 \right], \tag{9}$$

where $\tau > 0$ controls the amount of penalty. It is well-known (e.g., see the proof of Theorem 1 in Lounici (2014)) that $\widehat{\Sigma}_{2n}^\tau$ can be written explicitly as

$$\widehat{\Sigma}_*^\tau = \sum_{i=1}^d \max\left(\lambda_i\left(\widehat{\Sigma}_*\right) - \tau/2, 0\right) v_i(\widehat{\Sigma}_*)v_i(\widehat{\Sigma}_*)^T,$$

where $\lambda_i(\widehat{\Sigma}_*)$ and $v_i(\widehat{\Sigma}_*)$ are the eigenvalues and corresponding eigenvectors of $\widehat{\Sigma}_*$. We are ready to state the main result of this section.

**Theorem 3.1.** *For any $\tau \geq 36\sigma_0\sqrt{\frac{\beta}{m}}$,*

$$\left\|\widehat{\Sigma}_*^\tau - \Sigma_0\right\|_{\mathrm{F}}^2 \leq \inf_{A \in \mathbb{R}^{d \times d}} \left[ \|A - \Sigma_0\|_{\mathrm{F}}^2 + \frac{(1+\sqrt{2})^2}{8}\tau^2\mathrm{rank}(A) \right]. \tag{10}$$

*with probability $\geq 1 - 5d\log_2\left(\frac{2\sigma_{max}}{\sigma_{min}}\right)e^{-\beta}$.*

In particular, if $\mathrm{rank}(\Sigma_0) = r$ and $\tau = 36\sigma_0\sqrt{\frac{\beta}{m}}$, we obtain that

$$\left\|\widehat{\Sigma}_*^\tau - \Sigma_0\right\|_{\mathrm{F}}^2 \leq 162\,\sigma_0^2\left(1+\sqrt{2}\right)^2\frac{\beta r}{m}$$

with probability $\geq 1 - 5d\log_2\left(\frac{2\sigma_{\max}}{\sigma_{\min}}\right)e^{-\beta}$.

## 4 Proofs

### 4.1 Proof of Lemma 2.1

The result is a simple corollary of the following statement.

**Lemma 4.1.** *Set* $\theta = \frac{1}{\sigma}\sqrt{\frac{\beta}{m}}$, *where* $\sigma \geq \sigma_0$ *and* $m \geq \beta$. *Let* $\overline{d} := \sigma_0^2/\|\Sigma_0\|^2$. *Then, with probability at least* $1 - 5de^{-\beta}$,

$$\left\| \widehat{\Sigma} - \Sigma_0 \right\| \leq 2\sigma\sqrt{\frac{\beta}{m}}$$

$$+ C'\|\Sigma_0\| \left( \sqrt{\frac{\overline{d}\sigma}{\|\Sigma_0\|}} \left(\frac{\beta}{m}\right)^{\frac{3}{4}} + \frac{\sqrt{\overline{d}}\sigma}{\|\Sigma_0\|}\frac{\beta}{m} + \sqrt{\frac{\overline{d}\sigma}{\|\Sigma_0\|}}\left(\frac{\beta}{m}\right)^{\frac{5}{4}} + \overline{d}\left(\frac{\beta}{m}\right)^{\frac{3}{2}} + \frac{\overline{d}\beta^2}{m^2} + \overline{d}^{\frac{5}{4}}\left(\frac{\beta}{m}\right)^{\frac{9}{4}} \right),$$

*where* $C' > 1$ *is an absolute constant.*

Now, by Corollary **??** in the supplement, it follows that $\overline{d} = \sigma_0^2/\|\Sigma_0\|^2 \geq \operatorname{tr}(\Sigma_0)/\|\Sigma_0\| \geq 1$. Thus, assuming that the sample size satisfies $m \geq (6C')^4 \overline{d}\beta$, then, $\overline{d}\beta/m \leq 1/(6C')^4 < 1$, and by some algebraic manipulations we have that

$$\left\| \widehat{\Sigma} - \Sigma_0 \right\| \leq 2\sigma\sqrt{\frac{\beta}{m}} + \sigma\sqrt{\frac{\beta}{m}} = 3\sigma\sqrt{\frac{\beta}{m}}. \tag{11}$$

For completeness, a detailed computation is given in the supplement. This finishes the proof.

## 4.2 Proof of Lemma 4.1

Let $B_\beta = 11\sqrt{2\operatorname{tr}(\Sigma_0)\beta/m}$ be the error bound of the robust mean estimator $\widehat{\mu}$ defined in (3). Let $Z_i = X_i - \mu_0$, $\Sigma_\mu = \mathbb{E}\big[(Z_i - \mu)(Z_i - \mu)^T\big]$, $\forall i = 1, 2, \cdots, d$, and

$$\hat{\Sigma}_\mu = \frac{1}{m\theta}\sum_{i=1}^{m} \frac{(X_i - \mu)(X_i - \mu)^T}{\|X_i - \mu\|_2^2}\psi\left(\theta\|X_i - \mu\|_2^2\right),$$

for any $\|\mu\|_2 \leq B_\beta$. We begin by noting that the error can be bounded by the supremum of an empirical process indexed by $\mu$, i.e.

$$\left\| \hat{\Sigma} - \Sigma_0 \right\| \leq \sup_{\|\mu\|_2 \leq B_\beta} \left\| \hat{\Sigma}_\mu - \Sigma_0 \right\| \leq \sup_{\|\mu\|_2 \leq B_\beta} \left\| \hat{\Sigma}_\mu - \Sigma_\mu \right\| + \|\Sigma_\mu - \Sigma_0\| \tag{12}$$

with probability at least $1 - e^{-\beta}$. We first estimate the second term $\|\Sigma_\mu - \Sigma_0\|$. For any $\|\mu\|_2 \leq B_\beta$,

$$\|\Sigma_\mu - \Sigma_0\| = \left\| \mathbb{E}\big[(Z_i - \mu)(Z_i - \mu)^T - Z_i Z_i^T\big] \right\| = \sup_{\mathbf{v}\in\mathbb{R}^d:\|\mathbf{v}\|_2\leq 1} \left| \mathbb{E}\Big[\langle Z_i - \mu, \mathbf{v}\rangle^2 - \langle Z_i, \mathbf{v}\rangle^2\Big] \right|$$

$$= (\mu^T\mathbf{v})^2 \leq \|\mu\|_2^2 \leq B_\beta^2 = 242\frac{\operatorname{tr}(\Sigma_0)\beta}{m},$$

with probability at least $1 - e^{-\beta}$. It follows from Corollary **??** in the supplement that with the same probability

$$\|\Sigma_\mu - \Sigma_0\| \leq 242\frac{\sigma_0^2\beta}{\|\Sigma_0\|m} \leq 242\frac{\sigma^2\beta}{\|\Sigma_0\|m} = 242\|\Sigma_0\|\frac{\overline{d}\beta}{m}. \tag{13}$$

Our main task is then to bound the first term in (12). To this end, we rewrite it as a double supremum of an empirical process:

$$\sup_{\|\mu\|_2 \leq B_\beta} \left\| \hat{\Sigma}_\mu - \Sigma_\mu \right\| = \sup_{\|\mu\|_2 \leq B_\beta, \|\mathbf{v}\|_2 \leq 1} \left| \mathbf{v}^T\left(\hat{\Sigma}_\mu - \Sigma_\mu\right)\mathbf{v} \right|$$

It remains to estimate the supremum above.

**Lemma 4.2.** *Set* $\theta = \frac{1}{\sigma}\sqrt{\frac{\beta}{m}}$, *where* $\sigma \geq \sigma_0$ *and* $m \geq \beta$. *Let* $\overline{d} := \sigma_0^2/\|\Sigma_0\|^2$. *Then, with probability at least* $1 - 4de^{-\beta}$,

$$\sup_{\|\mu\|_2 \leq B_\beta, \|\mathbf{v}\|_2 \leq 1} \left| \mathbf{v}^T\left(\hat{\Sigma}_\mu - \Sigma_\mu\right)\mathbf{v} \right| \leq 2\sigma\sqrt{\frac{\beta}{m}}$$

$$+ C''\|\Sigma_0\| \left( \sqrt{\frac{\overline{d}\sigma}{\|\Sigma_0\|}} \left(\frac{\beta}{m}\right)^{\frac{3}{4}} + \frac{\sqrt{\overline{d}}\sigma}{\|\Sigma_0\|}\frac{\beta}{m} + \sqrt{\frac{\overline{d}\sigma}{\|\Sigma_0\|}}\left(\frac{\beta}{m}\right)^{\frac{5}{4}} + \overline{d}\left(\frac{\beta}{m}\right)^{\frac{3}{2}} + \frac{\overline{d}\beta^2}{m^2} + \overline{d}^{\frac{5}{4}}\left(\frac{\beta}{m}\right)^{\frac{9}{4}} \right),$$

*where* $C'' > 1$ *is an absolute constant.*

Note that $\sigma \geq \sigma_0$ by defnition, thus, $\overline{d} \leq \sigma^2/\|\Sigma_0\|^2$. Combining the above lemma with (12) and (13) finishes the proof.

## 4.3 Proof of Theorem 2.1

Define $\overline{j} := \min\{j \in \mathcal{J} : \sigma_j \geq \sigma_0\}$, and note that $\sigma_{\overline{j}} \leq 2\sigma_0$. We will demonstrate that $j_* \leq \overline{j}$ with high probability. Observe that

$$
\begin{aligned}
\Pr(j_* > \overline{j}) &\leq \Pr\left(\bigcup_{k \in \mathcal{J}:k > \overline{j}} \left\{\left\|\widehat{\Sigma}_{m,k} - \Sigma_{m,\overline{j}}\right\| > 6\sigma_k\sqrt{\frac{\beta}{n}}\right\}\right)\\
&\leq \Pr\left(\left\|\widehat{\Sigma}_{m,\overline{j}} - \Sigma_0\right\| > 3\sigma_{\overline{j}}\sqrt{\frac{\beta}{m}}\right) + \sum_{k \in \mathcal{J}: k > \overline{j}} \Pr\left(\left\|\widehat{\Sigma}_{m,k} - \Sigma_0\right\| > 3\sigma_k\sqrt{\frac{\beta}{m}}\right)\\
&\leq 5de^{-\beta} + 5d\log_2\left(\frac{\sigma_{\max}}{\sigma_{\min}}\right)e^{-\beta},
\end{aligned}
$$

where we applied (5) to estimate each of the probabilities in the sum under the assumption that the number of samples $m \geq C\overline{d}\beta$ and $\sigma_k \geq \sigma_{\overline{j}} \geq \sigma_0$. It is now easy to see that the event

$$
\mathcal{B} = \bigcap_{k \in \mathcal{J}:k \geq \overline{j}} \left\{\left\|\widehat{\Sigma}_{m,k} - \Sigma_0\right\| \leq 3\sigma_k\sqrt{\frac{\beta}{m}}\right\}
$$

of probability $\geq 1 - 5d\log_2\left(\frac{2\sigma_{\max}}{\sigma_{\min}}\right)e^{-\beta}$ is contained in $\mathcal{E} = \{j_* \leq \overline{j}\}$. Hence, on $\mathcal{B}$

$$
\left\|\widehat{\Sigma}_* - \Sigma_0\right\| \leq \|\widehat{\Sigma}_* - \widehat{\Sigma}_{m,\overline{j}}\| + \|\widehat{\Sigma}_{m,\overline{j}} - \Sigma_0\| \leq 6\sigma_{\overline{j}}\sqrt{\frac{\beta}{m}} + 3\sigma_{\overline{j}}\sqrt{\frac{\beta}{m}}
$$

$$
\leq 12\sigma_0\sqrt{\frac{\beta}{m}} + 6\sigma_0\sqrt{\frac{\beta}{m}} = 18\sigma_0\sqrt{\frac{\beta}{m}},
$$

and the claim follows.

## 4.4 Proof of Theorem 3.1

The proof is based on the following lemma:

**Lemma 4.3.** *Inequality (10) holds on the event* $\mathcal{E} = \left\{\tau \geq 2\left\|\widehat{\Sigma}_* - \Sigma_0\right\|\right\}$.

To verify this statement, it is enough to repeat the steps of the proof of Theorem 1 in Lounici (2014), replacing each occurrence of the sample covariance matrix by its "robust analogue" $\widehat{\Sigma}_*$.

It then follows from Theorem 2.1 that $\Pr(\mathcal{E}) \geq 1 - 5d\log_2\left(\frac{2\sigma_{\max}}{\sigma_{\min}}\right)e^{-\beta}$ whenever $\tau \geq 36\sigma_0\sqrt{\frac{\beta}{m}}$.

### Acknowledgments

Research of S. Minsker and X. Wei was partially supported by the National Science Foundation grant NSF DMS-1712956.

## Footnotes

[1] Balasubramanian and Yuan (2016)

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
