[Supplementary Material · supplement.pdf]

# Estimation of the covariance structure of heavy-tailed distributions

**Stanislav Minsker**
Department of Mathematics
University of Southern California
Los Angeles, CA 90007
minsker@usc.edu

**Xiaohan Wei**
Department of Electrical Engineering
University of Southern California
Los Angeles, CA 90007
xiaohanw@usc.edu

## 1 Supplement

### 1.1 Preliminaries

**Lemma 1.1.** *Consider any function $\phi : \mathbb{R} \to \mathbb{R}$ and $\theta > 0$. Suppose the following holds*

$$-\frac{1}{\theta} \log \left(1 - \theta x + \theta^2 x^2\right) \le \phi(x) \le \frac{1}{\theta} \log \left(1 + \theta x + \theta^2 x^2\right), \ \forall x \in \mathbb{R} \tag{1}$$

*then, we have for any matrix $A \in \mathbb{H}^{d \times d}$,*

$$-\frac{1}{\theta} \log \left(1 - \theta A + \theta^2 A^2\right) \le \phi(A) \le \frac{1}{\theta} \log \left(I + \theta A + \theta^2 A^2\right).$$

*Proof.* Note that for any $x \in \mathbb{R}$, $-\frac{1}{\theta} \log \left(1 - x\theta + x^2 \theta^2\right) \le \frac{1}{\theta} \log \left(1 + x\theta + x^2 \theta^2\right)$, then, the claim follows immediately from the definition of the matrix function. $\square$

The above lemma is useful in our context mainly due to the following lemma,

**Lemma 1.2.** *The truncation function $\frac{1}{\theta} \psi(\theta x) = sign(x) \cdot \left(|x| \wedge \frac{1}{\theta}\right)$ satisfies the assumption* (14) *in Lemma 5.1.*

*Proof.* Denote $f_1(x) = -\frac{1}{\theta} \log \left(1 - \theta x + \theta^2 x^2\right)$, $f_2(x) = \frac{1}{\theta} \log \left(1 + \theta x + \theta^2 x^2\right)$ and $g(x) = sign(x) \cdot \left(|x| \wedge \frac{1}{\theta}\right)$. Note first that

$$\begin{aligned}
f_1(0) = g(0) = f_2(0) = 0, \\
f_1(1/\theta) \le g(1/\theta) \le f_2(1/\theta), \\
f_1(-1/\theta) \le g(-1/\theta) \le f_2(-1/\theta),
\end{aligned}$$

and the subgradient

$$\partial g(x) = \begin{cases} 1, & x \in (-1/\theta, 1/\theta), \\ 0, & x \in (-\infty, -1/\theta) \cup (1/\theta, +\infty), \\ [0, 1], & x = -1/\theta, 1/\theta. \end{cases}$$

Next, we take the derivative of $f_2(x)$ and compare it to the derivative of $g(x)$.

$$f_2'(x) = \frac{1}{\theta} \cdot \frac{\theta + 2x\theta^2}{1 + x\theta + x^2\theta^2} = \frac{1 + 2x\theta}{1 + x\theta + x^2\theta^2}.$$

Note that $f_2'(x) \geq 1, x \in (0, 1/\theta)$, $f_2'(x) \geq 0, x \geq 1/\theta$, $f_2'(x) \leq 1, x \in (-1/\theta, 0]$ and $f_2'(x) \leq 0, x \leq -1/\theta$. Thus, we have $g(x) \leq f_2(x), \forall x \in \mathbb{R}$. Similarly, we can take the derivative of $f_1(x)$ and compare it to $g(x)$, which results in $f_1'(x) \leq 1, x \in (0, 1/\theta)$, $f_1'(x) \leq 0, x \geq 1/\theta$, $f_1'(x) \geq 1, x \in (-1/\theta, 0]$ and $f_2'(x) \geq 0, x \leq -1/\theta$. This implies $f_1(x) \leq g(x)$ and the Lemma is proved. $\qquad\square$

The following lemma demonstrates the importance of matrix logarithm function in matrix analysis, whose proof can be found in Bhatia (2013) and Tropp (2015),

**Lemma 1.3.** *(a) The matrix logarithm is operator monotone, that is, if $A \succ B \succ 0$ are two matrices in $\mathbb{H}^{d \times d}$, then, $\log(A) \succ \log(B)$.*
*(b) Given a fixed matrix $H \in \mathbb{H}^{d \times d}$, the function*

$$A \to tr \exp(H + \log(A))$$

*is concave on the cone of positive semi-definite matrices.*

The following lemma is a generalization of Chebyshev's association inequality. See Theorem 2.15 of Boucheron et al. (2013) for proof.

**Lemma 1.4** (FKG inequality). *Suppose $f, g : \mathbb{R}^d \to \mathbb{R}$ are two functions non-decreasing on each coordinate. Let $Y = [Y_1, Y_2, \cdots, Y_d]$ be a random vector taking values in $\mathbb{R}^d$, then,*

$$\mathbb{E}[f(X)g(X)] \geq \mathbb{E}[f(X)]\mathbb{E}[g(X)].$$

The following corollary follows immediately from the FKG inequality.

**Corollary 1.1.** *Let $Z = X - \mu_0$, then, we have $\sigma_0^2 = \|\mathbb{E}[ZZ^T \|Z\|_2^2]\| \geq tr\left(\mathbb{E}[ZZ^T]\right)\|\mathbb{E}[ZZ^T]\| = tr(\Sigma_0)\|\Sigma_0\|$.*

*Proof.* Consider any unit vector $\mathbf{v} \in \mathbb{R}^d$. It is enough to show $\mathbb{E}[(\mathbf{v}^T Z)^2 \|Z\|_2^2] \geq \mathbb{E}[(\mathbf{v}^T Z)^2]\mathbb{E}[\|Z\|_2^2]$. We change the coordinate by considering an orthonormal basis $\{\mathbf{v}_1, \cdots, \mathbf{v}_d\}$ with $\mathbf{v}_1 = \mathbf{v}$. Let $Y_i = \mathbf{v}_i^T Z, i = 1, 2, \cdots, d$, then we obtain,

$$\mathbb{E}[(\mathbf{v}^T Z)^2 \|Z\|_2^2] = \mathbb{E}[Y_1^2 \|Y\|_2^2] \geq \mathbb{E}[Y_1^2]\mathbb{E}[\|Y\|_2^2],$$

where the last inequality follows from FKG inequality by taking $f(Y_1^2, \cdots, Y_d^2) = Y_1^2$ and $g(Y_1^2, \cdots, Y_d^2) = \|Y\|_2^2$. $\qquad\square$

## 1.2 Additional computation in the proof of Lemma 2.1

In order to show (11), it is enough to show that

$$C'\|\Sigma_0\| \left( \sqrt{\frac{\overline{d}\sigma}{\|\Sigma_0\|}} \left(\frac{\beta}{m}\right)^{\frac{3}{4}} + \frac{\sqrt{\overline{d}}\sigma}{\|\Sigma_0\|}\frac{\beta}{m} + \sqrt{\frac{\overline{d}\sigma}{\|\Sigma_0\|}}\left(\frac{\beta}{m}\right)^{\frac{5}{4}} + \overline{d}\left(\frac{\beta}{m}\right)^{\frac{3}{2}} + \frac{\overline{d}\beta^2}{m^2} + \overline{d}^{\frac{5}{4}}\left(\frac{\beta}{m}\right)^{\frac{9}{4}} \right) \leq \sigma\sqrt{\frac{\beta}{m}}.$$

Note that $\overline{d} = \sigma_0^2/\|\Sigma_0\|^2 \geq tr(\Sigma_0)/\|\Sigma_0\| \geq 1$, and assuming that the sample size satisfies $m \geq (6C')^4 \overline{d}\beta$, we have $\overline{d}\beta/m \leq 1/(6C')^4 < 1$. We then bound each of the 6 terms on the left side.

$$C'\|\Sigma_0\|\sqrt{\frac{\overline{d}\sigma}{\|\Sigma_0\|}}\left(\frac{\beta}{m}\right)^{\frac{3}{4}} = C'\sqrt{\sigma}\left(\frac{\beta}{m}\right)^{\frac{1}{4}} \cdot \left(\frac{\|\Sigma_0\|\overline{d}\beta}{m}\right)^{1/4} \cdot \left(\frac{\|\Sigma_0\|\overline{d}\beta}{m}\right)^{1/4}$$

$$\leq C'\sqrt{\sigma}\left(\frac{\beta}{m}\right)^{\frac{1}{4}} \cdot \left(\frac{\|\Sigma_0\|\overline{d}\beta}{m}\right)^{1/4} \cdot \frac{1}{6C'}$$

$$= \frac{1}{6}\sqrt{\sigma\sigma_0}\sqrt{\frac{\beta}{m}} \leq \frac{1}{6}\sigma\sqrt{\frac{\beta}{m}},$$

$$C'\|\Sigma_0\| \cdot \sqrt{\overline{d}}\frac{\sigma}{\|\Sigma_0\|}\frac{\beta}{m} = C'\sigma\sqrt{\frac{\beta}{m}} \cdot \sqrt{\frac{\overline{d}\beta}{m}} \leq C'\sigma\sqrt{\frac{\beta}{m}}\frac{1}{(6C')^2} \leq \frac{1}{6}\sigma\sqrt{\frac{\beta}{m}},$$

$$C'\|\Sigma_0\|\sqrt{\frac{\overline{d}\sigma}{\|\Sigma_0\|}}\left(\frac{\beta}{m}\right)^{\frac{5}{4}} \leq C'\|\Sigma_0\|\sqrt{\frac{\overline{d}\sigma}{\|\Sigma_0\|}}\left(\frac{\beta}{m}\right)^{\frac{3}{4}} \leq \frac{1}{6}\sigma\sqrt{\frac{\beta}{m}}.$$

Note that we have the following

$$C'\|\Sigma_0\|\overline{d}\frac{\beta}{m} = C'\|\Sigma_0\|\left(\frac{\overline{d}\beta}{m}\right)^{\frac{1}{2}}\left(\frac{\overline{d}\beta}{m}\right)^{\frac{1}{2}} \le C'\|\Sigma_0\|\left(\frac{\overline{d}\beta}{m}\right)^{\frac{1}{2}}\frac{1}{(6C')^2} \le \frac{1}{6}\sigma_0\sqrt{\frac{\beta}{m}} \le \frac{1}{6}\sigma\sqrt{\frac{\beta}{m}},$$

thus, the rest three terms can be bounded as follows,

$$C'\|\Sigma_0\|\overline{d}\left(\frac{\beta}{m}\right)^{\frac{3}{2}} \le C'\|\Sigma_0\|\overline{d}\frac{\beta}{m} \le \frac{1}{6}\sigma\sqrt{\frac{\beta}{m}}$$

$$C'\|\Sigma_0\|\overline{d}\frac{\beta^2}{m^2} \le C'\|\Sigma_0\|\overline{d}\frac{\beta}{m} \le \frac{1}{6}\sigma\sqrt{\frac{\beta}{m}}$$

$$C'\|\Sigma_0\|\overline{d}^{\frac{5}{4}}\left(\frac{\beta}{m}\right)^{\frac{9}{4}} \le C'\|\Sigma_0\|\overline{d}^{\frac{5}{4}}\left(\frac{\beta}{m}\right)^{\frac{5}{4}} \le C'\|\Sigma_0\|\overline{d}\frac{\beta}{m} \le \frac{1}{6}\sigma\sqrt{\frac{\beta}{m}}.$$

Overall, we have (11) holds.

### 1.3 Proof of Lemma 4.2

First of all, by definition of $\widehat{\Sigma}_\mu$, we have

$$\sup_{\|\mu\|_2 \le B_\beta, \|\mathbf{v}\|_2 \le 1}\left|\mathbf{v}^T(\hat{\Sigma}_\mu - \Sigma_\mu)\mathbf{v}\right| = \sup_{\|\mu\|_2 \le B_\beta, \|\mathbf{v}\|_2 \le 1}\left|\frac{1}{m\theta}\sum_{i=1}^{m}\langle Z_i - \mu, \mathbf{v}\rangle^2\frac{\psi\left(\theta\|Z_i - \mu\|_2^2\right)}{\|Z_i - \mu\|_2^2} - \mathbb{E}\left[\langle Z_i - \mu, \mathbf{v}\rangle^2\right]\right|.$$

Expanding the squares on the right hand side gives

$$\sup_{\|\mu\|_2 \le B_\beta}\left\|\hat{\Sigma}_\mu - \Sigma_\mu\right\| \le \sup_{\|\mu\|_2 \le B_\beta, \|\mathbf{v}\|_2 \le 1}\left|\frac{1}{m}\sum_{i=1}^{m}\langle Z_i, \mathbf{v}\rangle^2\frac{\psi\left(\theta\|Z_i - \mu\|_2^2\right)}{\theta\|Z_i - \mu\|_2^2} - \mathbb{E}\left[\langle Z_i, \mathbf{v}\rangle^2\right]\right| \quad \text{(I)}$$

$$+ 2\sup_{\|\mu\|_2 \le B_\beta, \|\mathbf{v}\|_2 \le 1}\left|\frac{1}{m}\sum_{i=1}^{m}\langle Z_i, \mathbf{v}\rangle\langle\mu, \mathbf{v}\rangle\frac{\psi\left(\theta\|Z_i - \mu\|_2^2\right)}{\theta\|Z_i - \mu\|_2^2} - \mathbb{E}[\langle Z_i, \mathbf{v}\rangle\langle\mu, \mathbf{v}\rangle]\right| \quad \text{(II)}$$

$$+ \sup_{\|\mu\|_2 \le B_\beta, \|\mathbf{v}\|_2 \le 1}\left|\frac{1}{m}\sum_{i=1}^{m}\langle\mu, \mathbf{v}\rangle^2\frac{\psi\left(\theta\|Z_i - \mu\|_2^2\right)}{\theta\|Z_i - \mu\|_2^2} - \langle\mu, \mathbf{v}\rangle^2\right|. \quad \text{(III)}$$

We will then bound these three terms separately. Note that given $\|\widehat{\mu} - \mu_0\|_2 \le B_\beta$, the term (III) can be readily bounded as follows using the fact that $0 \le \psi(x) \le x, \forall x \ge 0$,

$$\text{(III)} = \sup_{\|\mu\|_2 \le B_\beta, \|\mathbf{v}\|_2 \le 1}\left|\langle\mu, \mathbf{v}\rangle^2\left(\frac{1}{m}\sum_{i=1}^{m}\frac{\psi\left(\theta\|Z_i - \mu\|_2^2\right)}{\theta\|Z_i - \mu\|_2^2} - 1\right)\right| \le \sup_{\|\mu\|_2 \le B_\beta, \|\mathbf{v}\|_2 \le 1}\langle\mu, \mathbf{v}\rangle^2 \le B_\beta^2$$

$$= 242\frac{tr(\Sigma_0)}{m}\beta \le 242\frac{\sigma_0^2\beta}{\|\Sigma_0\|m} \le 242\|\Sigma_0\|\frac{\overline{d}\beta}{m}, \quad (2)$$

where the second from the last inequality follows from Corollary 5.1 and the last inequality follows from $\overline{d} = \sigma_0^2/\|\Sigma_0\|^2$.

The rest two terms are bounded through the following lemma whose proof is delayed to the next section:

**Lemma 1.5.** *Given $\|\widehat{\mu} - \mu_0\|_2 \le B_\beta$, with probability at least $1 - 4de^{-\beta}$, we have the following two bounds hold,*

$$\text{(I)} \le 2\sigma\sqrt{\frac{\beta}{m}} + 22\|\Sigma_0\|\left(\sqrt{2}\overline{d}^{\frac{1}{4}}\left(\frac{\beta}{m}\right)^{\frac{3}{4}} + 2\sqrt{2}\sqrt{\frac{\overline{d}\sigma}{\|\Sigma_0\|}}\left(\frac{\beta}{m}\right)^{\frac{5}{4}} + 11\overline{d}^{\frac{1}{2}}\left(\frac{\beta}{m}\right)^{\frac{3}{2}} + 22\frac{\overline{d}\beta^2}{m^2}\right),$$

$$\text{(II)} \le 11\|\Sigma_0\|\left(\sqrt{2}\sqrt{\frac{\overline{d}\sigma}{\|\Sigma_0\|}}\left(\frac{\beta}{m}\right)^{\frac{3}{4}} + 3\sqrt{2}\sqrt{\overline{d}}\frac{\sigma}{\|\Sigma_0\|}\frac{\beta}{m} + 44\overline{d}^{\frac{3}{4}}\left(\frac{\beta}{m}\right)^{\frac{5}{4}}\right.$$

$$\left. + 44\sqrt{2\widetilde{d}}\left(\frac{\beta}{m}\right)^{\frac{3}{2}} + 242\sqrt{2}\frac{\overline{d}\beta^2}{m^2} + 484\overline{d}^{\frac{5}{4}}\left(\frac{\beta}{m}\right)^{\frac{9}{4}}\right).$$

Note that since $\sigma \geq \sigma_0$, we have $\sigma/\|\Sigma_0\| \geq \sigma_0/\|\Sigma_0\| = \sqrt{d}$. Combining the above lemma with (15) finishes the proof of Lemma 4.2.

## 1.4 Proof of Lemma 5.5

Before proving the Lemma, we introduce the following abbreviations:

$$g_{\mathbf{v}}(Z_i) = \langle Z_i, \mathbf{v} \rangle^2 \frac{\psi\left(\theta\|Z_i\|_2^2\right)}{\theta\|Z_i\|_2^2}, \quad h_\mu(Z_i) = \frac{\|Z_i\|_2^2}{\psi\left(\theta\|Z_i\|_2^2\right)} \frac{\psi\left(\theta\|Z_i - \mu\|_2^2\right)}{\|Z_i - \mu\|_2^2},$$

$$\tilde{g}_{\mathbf{v}}(Z_i) = \langle Z_i, \mathbf{v} \rangle \frac{\psi\left(\theta\|Z_i\|_2^2\right)}{\theta\|Z_i\|_2^2}.$$

Our analysis relies on the following simply yet important fact which gives deterministic upper and lower bound of $h_\mu(Z_i)$ around 1. Its proof is delayed to the next section.

**Lemma 1.6.** *For any $\mu$ such that $\|\mu\|_2 \leq B_\beta$, the following holds:*

$$1 - 2B_\beta\sqrt{\theta} - B_\beta^2\theta \leq h_\mu(Z_i) \leq 1 + 2B_\beta\sqrt{\theta} + B_\beta^2\theta.$$

The following Lemma gives a general concentration bound for heavy tailed random matrices under a mapping $\phi(\cdot)$.

**Lemma 1.7.** *Let $A_1, A_2, \cdots, A_m$ be a sequence of i.i.d. random matrices in $\mathbb{H}^{d \times d}$ with zero mean and finite second moment $\sigma_A^2 = \|\mathbb{E}[A_i^2]\|$. Let $\phi(\cdot)$ be any function satisfying the assumption (14) of Lemma 5.1. Then, for any $t > 0$,*

$$\Pr\left(\sum_{i=1}^m (\phi(A_i) - \mathbb{E}[A_i]) \geq t\sqrt{m}\right) \leq 2d\exp\left(-t\theta\sqrt{m} + m\theta^2\sigma_A^2\right).$$

*Specifically, if the assumption (14) holds for $\theta = \frac{t}{2\sqrt{m}\sigma_A}$, then we obtain the subgaussian tail $2d\exp(-t^2/4\sigma_A^2)$.*

The intuition behind this lemma is that the $\log(1 + x)$ tends to "robustify" a random variable by implicitly trading the bias for a tight concentration. A scalar version of such lemma with a similar idea is first introduced in the seminal work Catoni (2012). The proof of the current matrix version is similar to Lemma 3.1 and Theorem 3.1 of Minsker (2016) by modifying only the constants. We omitted the details here for brevity. Note that this lemma is useful in our context by choosing $\phi(x) = \frac{1}{\theta}\psi(\theta x)$. Next, we prove two parts of Lemma 5.5 separately.

*Proof of (I) in Lemma 5.5.* Using the abbreviation introduced at the beginning of this section, we have

$$(I) = \sup_{\|\mu\|_2 \leq B_\beta, \|\mathbf{v}\|_2 \leq 1} \left| \frac{1}{m}\sum_{i=1}^m g_{\mathbf{v}}(Z_i)h_\mu(Z_i) - \mathbb{E}\left[\langle Z_i, \mathbf{v} \rangle^2\right] \right|$$

We further split it into two terms as follows:

$$(I) \leq \sup_{\|\mu\|_2 \leq B_\beta, \|\mathbf{v}\|_2 \leq 1} \left| \frac{1}{m}\sum_{i=1}^m g_{\mathbf{v}}(Z_i)\left(h_\mu(Z_i) - 1\right) \right| + \sup_{\|\mathbf{v}\| \leq} \left| \frac{1}{m}\sum_{i=1}^m g_{\mathbf{v}}(Z_i) - \mathbb{E}\left[\langle Z_i, \mathbf{v} \rangle^2\right] \right| \quad (3)$$

The two terms in (16) are bounded as follows:

1. For the second term in (16), note that we can write it back into the matrix form as

$$\left\| \frac{1}{m\theta}\sum_{i=1}^m Z_iZ_i^T \frac{\psi\left(\theta\|Z_i\|_2^2\right)}{\|Z_i\|_2^2} - \mathbb{E}\left[Z_iZ_i^T\right] \right\|.$$

Note that the matrix $Z_iZ_i^T$ is a rank one matrix with the eigenvalue equal to $\|Z_i\|_2^2$, so it follows from the definition of matrix function,

$$Z_iZ_i^T \frac{\psi\left(\theta\|Z_i\|_2^2\right)}{\|Z_i\|_2^2} = \frac{1}{\theta}\psi\left(\theta Z_iZ_i^T\right).$$

Now, applying Lemma 5.2 setting $\theta = \frac{t}{2\sigma^2 \sqrt{m}}$ together with Lemma 5.7 gives

$$\Pr\left(\left\|\frac{1}{m\theta}\sum_{i=1}^{m} Z_i Z_i^T \frac{\psi\left(\theta\|Z_i\|_2^2\right)}{\|Z_i\|_2^2} - \mathbb{E}\left[Z_i Z_i^T\right]\right\| \geq t/\sqrt{m}\right) \leq 2d\exp(-t^2/4\sigma^2).$$

Setting $t = 2\sigma\sqrt{\beta}$ (which results in $\theta = \frac{1}{\sigma}\sqrt{\frac{\beta}{m}}$) gives

$$\left\|\frac{1}{m\theta}\sum_{i=1}^{m} Z_i Z_i^T \frac{\psi\left(\theta\|Z_i\|_2^2\right)}{\|Z_i\|_2^2} - \mathbb{E}\left[Z_i Z_i^T\right]\right\| \leq 2\sigma\sqrt{\frac{\beta}{m}} \tag{4}$$

with probability at least $1 - 2de^{-\beta}$.

2. For the first term in (16), by the fact that $g_{\mathbf{v}}(Z_i) \geq 0$ and Lemma 5.6,

$$\sup_{\|\mu\|_2 \leq B_\beta, \|\mathbf{v}\|_2 \leq 1}\left|\frac{1}{m}\sum_{i=1}^{m} g_{\mathbf{v}}(Z_i)\left(h_\mu(Z_i) - 1\right)\right|$$

$$\leq \sup_{\|\mu\|_2 \leq B_\beta, \|\mathbf{v}\|_2 \leq 1}\frac{1}{m}\sum_{i=1}^{m} g_{\mathbf{v}}(Z_i)\left|h_\mu(Z_i) - 1\right|$$

$$\leq \sup_{\|\mathbf{v}\|_2 \leq 1}\frac{1}{m}\sum_{i=1}^{m} g_{\mathbf{v}}(Z_i)\left(2B_\beta\sqrt{\theta} + B_\beta^2\theta\right)$$

$$\leq \left(\left\|\mathbb{E}\left[Z_i Z_i^T\right]\right\| + 2\sigma\sqrt{\frac{\beta}{m}}\right)\left(2B_\beta\sqrt{\theta} + B_\beta^2\theta\right),$$

with probability at least $1 - 2de^{-\beta}$, where the last inequality follows from the same argument leading to (17). Note that $\mathbb{E}\left[Z_i Z_i^T\right] = \Sigma_0$.

Overall, we get

$$(\text{I}) \leq 2\sigma\sqrt{\frac{\beta}{m}} + \left(\|\Sigma_0\| + 2\sigma\sqrt{\frac{\beta}{m}}\right)\left(2B_\beta\sqrt{\theta} + B_\beta^2\theta\right),$$

with probability at least $1 - 2de^{-\beta}$. Now we substitute $B_\beta = 11\sqrt{2\text{tr}(\Sigma_0)\beta/m}$ and $\theta = \frac{1}{\sigma}\sqrt{\frac{\beta}{m}}$ into the above bound gives

$$(\text{I}) \leq 2\sigma\sqrt{\frac{\beta}{m}} + 22\sqrt{2}\|\Sigma_0\|\sqrt{\frac{\text{tr}(\Sigma_0)}{\sigma}}\left(\frac{\beta}{m}\right)^{\frac{3}{4}} + 242\|\Sigma_0\|\frac{\text{tr}\Sigma_0}{\sigma}\left(\frac{\beta}{m}\right)^{\frac{3}{2}}$$

$$+ 44\sqrt{2}\sqrt{\sigma\text{tr}(\Sigma_0)}\left(\frac{\beta}{m}\right)^{\frac{5}{4}} + 484\text{tr}(\Sigma_0)\left(\frac{\beta}{m}\right)^2$$

Using Corollary 5.1, we have

$$\frac{\text{tr}(\Sigma_0)}{\sigma} \leq \frac{\text{tr}(\Sigma_0)}{\sigma_0} \leq \frac{\text{tr}(\Sigma_0)}{\sqrt{\text{tr}(\Sigma_0)\|\Sigma_0\|}} \leq \frac{\sigma_0}{\|\Sigma_0\|} \leq \overline{d}, \tag{5}$$

and also,

$$\text{tr}(\Sigma_0) \leq \|\Sigma_0\|\sigma_0^2/\|\Sigma_0\|^2 \leq \|\Sigma_0\|\overline{d}. \tag{6}$$

Substitute these two bounds into the bound of (I) gives the final bound for (I) stated in Lemma 5.5 with probability at least $1 - 2de^{-\beta}$. $\qquad\square$

*Proof of (II) in Lemma 5.5.* First of all, using the definition of $\tilde{g}_{\mathbf{v}}(Z_i)$ and $h_\mu(Z_i)$, we can rewrite (II) as follows:

$$(\text{II}) = \sup_{\|\mu\|_2 \leq B_\beta, \|\mathbf{v}\|_2 \leq 1}\left|\frac{1}{m}\sum_{i=1}^{m} \tilde{g}_{\mathbf{v}}(Z_i)h_\mu(Z_i)\langle\mu, \mathbf{v}\rangle - \mathbb{E}[\langle Z_i, \mathbf{v}\rangle]\langle\mu, \mathbf{v}\rangle\right|$$

$$\leq B_\beta \cdot \sup_{\|\mu\|_2 \leq B_\beta, \|\mathbf{v}\|_2 \leq 1}\left|\frac{1}{m}\sum_{i=1}^{m} \tilde{g}_{\mathbf{v}}(Z_i)h_\mu(Z_i) - \mathbb{E}[\langle Z_i, \mathbf{v}\rangle]\right|.$$

Similar to the analysis of (I), we further split the above term into two terms and get

$$(II) \leq \underbrace{B_\beta \sup_{\|\mu\|_2 \leq B_\beta, \|\mathbf{v}\|_2 \leq 1} \left| \frac{1}{m} \sum_{i=1}^m \tilde{g}_\mathbf{v}(Z_i) \left( h_\mu(Z_i) - 1 \right) \right|}_{(IV)} + \underbrace{B_\beta \sup_{\|\mathbf{v}\|_2 \leq 1} \left| \frac{1}{m} \sum_{i=1}^m \tilde{g}_\mathbf{v}(Z_i) - \mathbb{E}[\langle Z_i, \mathbf{v} \rangle] \right|}_{(V)}.$$

(7)

For the first term, by Cauchy-Schwarz inequality and then Lemma 5.6, we get

$$\begin{aligned}
(IV) \leq & B_\beta \sup_{\|\mu\|_2 \leq B_\beta, \|\mathbf{v}\|_2 \leq 1} \frac{1}{m} \sum_{i=1}^m |\tilde{g}_\mathbf{v}(Z_i) \left( h_\mu(Z_i) - 1 \right)| \\
\leq & B_\beta \sup_{\|\mu\|_2 \leq B_\beta, \|\mathbf{v}\|_2 \leq 1} \left( \frac{1}{m} \sum_{i=1}^m \tilde{g}_\mathbf{v}(Z_i)^2 \right)^{1/2} \left( \frac{1}{m} \sum_{i=1}^m |h_\mu(Z_i) - 1|^2 \right)^{1/2} \\
\leq & B_\beta \sup_{\|\mathbf{v}\|_2 \leq 1} \left( \frac{1}{m} \sum_{i=1}^m \tilde{g}_\mathbf{v}(Z_i)^2 \right)^{1/2} \left( 2B_\beta \sqrt{\theta} + B_\beta^2 \theta \right).
\end{aligned}$$

Note that $\frac{1}{\theta} \psi \left( \theta \|Z_i\|_2^2 \right) / \|Z_i\|_2^2 \leq 1$, then, it follows,

$$\tilde{g}_\mathbf{v}(Z_i)^2 = \langle Z_i, \mathbf{v} \rangle^2 \left( \frac{\frac{1}{\theta} \psi \left( \theta \|Z_i\|_2^2 \right)}{\|Z_i\|_2^2} \right)^2 \leq \langle Z_i, \mathbf{v} \rangle^2 \frac{\frac{1}{\theta} \psi \left( \theta \|Z_i\|_2^2 \right)}{\|Z_i\|_2^2}.$$

Thus, by the same analysis leading to (17), we get

$$(IV) \leq B_\beta \left( \|\mathbb{E}[Z_i Z_i^T]\| + 2\sigma \sqrt{\frac{\beta}{m}} \right)^{1/2} \left( 2B_\beta \sqrt{\theta} + B_\beta^2 \theta \right),$$

(8)

with probability at least $1 - 2de^{-\beta}$. For the second term (V), notice that $\mathbb{E}[Z_i] = 0$, thus we have

$$\begin{aligned}
(V) \leq & B_\beta \sup_{\|\mathbf{v}\|_2 \leq 1} \left| \left\langle \frac{1}{m} \sum_{i=1}^m \frac{Z_i}{\|Z_i\|_2^2} \frac{1}{\theta} \psi(\theta \|Z_i\|_2^2), \mathbf{v} \right\rangle \right| \leq B_\beta \left\| \frac{1}{m} \sum_{i=1}^m \frac{Z_i}{\|Z_i\|_2^2} \|Z_i\|_2^2 \wedge \frac{1}{\theta} \right\|_2 \\
\leq & B_\beta \left\| \frac{1}{m} \sum_{i=1}^m \frac{Z_i}{\|Z_i\|_2^2} \|Z_i\|_2^2 \wedge \frac{1}{\theta} - \mathbb{E}\left[ \frac{Z_i}{\|Z_i\|_2^2} \|Z_i\|_2^2 \wedge \frac{1}{\theta} \right] \right\|_2 + B_\beta \left\| \mathbb{E}\left[ \frac{Z_i}{\|Z_i\|_2^2} \|Z_i\|_2^2 \wedge \frac{1}{\theta} \right] \right\|_2.
\end{aligned}$$

(9)

For the second term, which measures the bias, we have by the fact $\mathbb{E}[Z_i] = 0$,

$$\begin{aligned}
\left\| \mathbb{E}\left[ \frac{Z_i}{\|Z_i\|_2^2} \|Z_i\|_2^2 \wedge \frac{1}{\theta} \right] \right\|_2 = & \left\| \mathbb{E}\left[ Z_i \left( \frac{\|Z_i\|_2^2 \wedge \frac{1}{\theta}}{\|Z_i\|_2^2} - 1 \right) \right] \right\|_2 = \sup_{\|\mathbf{v}\|_2 \leq 1} \mathbb{E}\left[ \langle Z_i, \mathbf{v} \rangle \left( \frac{\|Z_i\|_2^2 \wedge \frac{1}{\theta}}{\|Z_i\|_2^2} - 1 \right) \right] \\
& \leq \sup_{\|\mathbf{v}\|_2 \leq 1} \mathbb{E}\left[ \langle Z_i, \mathbf{v} \rangle 1_{\{\|Z_i\|_2 \geq 1/\sqrt{\theta}\}} \right].
\end{aligned}$$

Now by Cauchy-Schwarz inequality and then Markov inequality, we obtain,

$$\begin{aligned}
\sup_{\|\mathbf{v}\|_2 \leq 1} \mathbb{E}\left[ \langle Z_i, \mathbf{v} \rangle 1_{\{\|Z_i\|_2 \geq 1/\sqrt{\theta}\}} \right] \leq & \sqrt{\sup_{\|\mathbf{v}\|_2 \leq 1} \mathbb{E}\left[ \langle Z_i, \mathbf{v} \rangle^2 \right]} Pr(\|Z_i\|_2 \geq 1/\sqrt{\theta})^{1/2} \leq \sqrt{\|\Sigma_0\|} \mathbb{E}\left[ \|Z_i\|_2^2 \right]^{1/2} \sqrt{\theta} \\
= & \sqrt{\|\Sigma_0\|} \frac{tr(\Sigma_0)^{1/2} \beta^{1/4}}{m^{1/4} \sigma^{1/2}} \leq \frac{(\|\Sigma_0\| tr(\Sigma_0))^{1/4} \beta^{1/4}}{m^{1/4}} \leq \left( \frac{\sigma^2}{m} \beta \right)^{1/4},
\end{aligned}$$

where the last two inequalities both follow from Lemma 5.1. This gives the second term in (22) is given by $B_\beta \left( \frac{\sigma^2}{m} \beta \right)^{1/4}$.

For the first term in (22), note that for any vector $\mathbf{x} \in \mathbb{R}^d$,

$$\|\mathbf{x}\|_2 = \left\| \begin{bmatrix} 0 & \mathbf{x}^T \\ \mathbf{x} & 0 \end{bmatrix} \right\|,$$

and furthermore, the matrix $\begin{bmatrix} 0 & \mathbf{x}^T \\ \mathbf{x} & 0 \end{bmatrix}$ has two same eigenvalues equal to $\|\mathbf{x}\|_2$, which follows from

$$\begin{bmatrix} 0 & \mathbf{x}^T \\ \mathbf{x} & 0 \end{bmatrix}^2 = \begin{bmatrix} \|\mathbf{x}\|_2^2 & 0 \\ 0 & \mathbf{x}\mathbf{x}^T \end{bmatrix}.$$

Thus, if we take

$$A_i = \begin{bmatrix} 0 & Z_i^T \\ Z_i & 0 \end{bmatrix} \frac{\|Z_i\|_2^2 \wedge \frac{1}{\theta}}{\|Z_i\|_2^2},$$

Then, the first term of (22) is equal to $\left\| \frac{1}{m} \sum_{i=1}^m A_i - \mathbb{E}[A_i] \right\|$. For this $A_i$, we have

$$\|\mathbb{E}[A_i^2]\| \leq \mathbb{E}[\|Z_i\|_2^2] = tr(\Sigma_0), \quad \|A_i\| \leq \frac{1}{\sqrt{\theta}} = \frac{m^{1/4}\sigma^{1/2}}{\beta^{1/4}}.$$

By matrix Bernstein's inequality (Tropp (2012)), we obtain the bound

$$\Pr\left( \left\| \frac{1}{m} \sum_{i=1}^m A_i - \mathbb{E}[A_i] \right\| \geq t \right) \leq d \exp\left( -\frac{3}{8} \left( \frac{mt^2}{\sigma^2} \wedge m\sqrt{\theta}t \right) \right) = d \exp\left( -\frac{3}{8} \left( \frac{mt^2}{\sigma^2} \wedge \frac{m^{3/4}\beta^{1/4}t}{\sigma^{1/2}} \right) \right),$$

where $c$ is a fixed positive constant. Taking $t = 3\sqrt{\frac{\sigma^2\beta}{\|\Sigma_0\|m}}$ gives

$$\Pr\left( \left\| \frac{1}{m} \sum_{i=1}^m A_i - \mathbb{E}[A_i] \right\| \geq 3\sqrt{\frac{\sigma^2}{m}\beta} \right) \leq d \exp\left( -3\beta \wedge \left( m^{1/4}\beta^{3/4}\overline{d}^{-1/4} \right) \right) \leq d \exp(-\beta),$$

where $\overline{d} = \sigma^2/\|\Sigma_0\|^2 \geq \sigma_0^2/\|\Sigma_0\|^2 \geq tr(\Sigma_0)/\|\Sigma_0\| \geq 1$ and the last inequality follows from the assumption that $m \geq \beta$. Overall, term (V) is bounded as follows

$$(\text{V}) \leq B_\beta \left( \frac{\sigma^2}{m}\beta \right)^{1/4} + 3B_\beta \sqrt{\frac{\sigma^2\beta}{\|\Sigma_0\|m}},$$

with probability at least $1 - de^{-\beta}$. Note that $\mathbb{E}[Z_i Z_i^T] = \Sigma_0$, then, combining with (21), the term (II) is bounded as

$$(\text{II}) \leq B_\beta \left( \|\Sigma_0\|^{\frac{1}{2}} + \sqrt{2}\sigma^{\frac{1}{2}} \left( \frac{\beta}{m} \right)^{\frac{1}{4}} \right) \left( 2B_\beta\sqrt{\theta} + B_\beta^2\theta \right) + B_\beta \left( \frac{\sigma^2}{m}\beta \right)^{1/4} + 3B_\beta \sqrt{\frac{\sigma^2\beta}{\|\Sigma_0\|m}},$$

with probability at least $1 - 2de^{-\beta}$. Substituting $B_\beta = 11\sqrt{\frac{2tr(\Sigma_0)\beta}{m}}$ and $\theta = \frac{1}{\sigma}\sqrt{\frac{\beta}{m}}$ gives

$$(\text{II}) \leq 11\sqrt{2}\sqrt{tr(\Sigma_0)\sigma} \left( \frac{\beta}{m} \right)^{\frac{3}{4}} + 33\sqrt{2}\frac{\sqrt{tr(\Sigma_0)}\sigma}{\|\Sigma_0\|^{1/2}}\frac{\beta}{m} + 484\|\Sigma_0\|^{1/2}\frac{tr(\Sigma_0)}{\sigma^{1/2}} \left( \frac{\beta}{m} \right)^{\frac{5}{4}}$$

$$+ 484\sqrt{2}tr(\Sigma_0) \left( \frac{\beta}{m} \right)^{\frac{3}{2}} + 2\sqrt{2} \cdot 11^3 \|\Sigma_0\|^{\frac{1}{2}}\frac{tr(\Sigma_0)^{3/2}}{\sigma} \left( \frac{\beta}{m} \right)^2 + 4 \cdot 11^3 \frac{tr(\Sigma_0)^{3/2}}{\sigma^{1/2}} \left( \frac{\beta}{m} \right)^{\frac{9}{4}}.$$

Using the bounds (18) and (19) with some algebraic manipulations, we have the second bound in Lemma 5.5 holds with probability at least $1 - 2de^{-\beta}$. $\qquad\square$

## 1.5 Proof of Lemma 5.6

We divide our analysis into the following four cases:

1. If $\|Z_i\|_2^2 \leq 1/\theta$ and $\|Z_i - \mu\|_2^2 \leq 1/\theta$, then, we have $h_\mu(Z_i) = 1$.

2. If $\|Z_i\|_2^2 \leq 1/\theta$ and $\|Z_i - \mu\|_2^2 > 1/\theta$. Since $\|\mu\| \leq B_\beta$, it follows $\|Z_i - \mu\|_2 \leq \sqrt{1/\theta} + B_\beta$, and we have

$$h_\mu(Z_i) = \frac{1/\theta}{\|Z_i - \mu\|_2^2} \leq 1,$$

$$h_\mu(Z_i) \geq \frac{1/\theta}{\left(\sqrt{1/\theta} + B_\beta\right)^2} = \frac{1}{1 + 2B_\beta\sqrt{\theta} + B_\beta^2\theta}$$

$$\geq 1 - 2B_\beta\sqrt{\theta} - B_\beta^2\theta,$$

where the last inequality follows from the fact $\frac{1}{1+x} \geq 1 - x, \ \forall x \geq 0$.

3. If $\|Z_i\|_2^2 > 1/\theta$ and $\|Z_i - \mu\|_2^2 \leq 1/\theta$. Since $\|\mu\|_2 \leq B_\beta$, it follows $\|Z_i\|_2 \leq \sqrt{1/\theta} + B_\beta$, and we have

$$h_\mu(Z_i) = \frac{\|Z_i\|_2^2}{1/\theta} \geq 1,$$

$$h_\mu(Z_i) \leq \frac{\left(\sqrt{1/\theta} + B_\beta\right)^2}{1/\theta} = 1 + 2B_\beta\sqrt{\theta} + B_\beta^2\theta.$$

4. If $\|Z_i\|_2^2 > 1/\theta$ and $\|Z_i - \mu\|_2^2 > 1/\theta$. Then, we have

$$h_\mu(Z_i) = \frac{\|Z_i\|_2^2}{\|Z_i - \mu\|_2^2} \leq \frac{(\|Z_i - \mu\|_2 + B_\beta)^2}{\|Z_i - \mu\|_2^2}$$

$$\leq \left(\frac{1/\sqrt{\theta} + B_\beta}{1/\sqrt{\theta}}\right)^2 \leq 1 + 2B_\beta\sqrt{\theta} + B_\beta^2\theta,$$

$$h_\mu(Z_i) \geq \frac{\|Z_i\|_2^2}{(\|Z_i\|_2 + B_\beta)^2} \geq \left(\frac{1/\sqrt{\theta}}{1/\sqrt{\theta} + B_\beta}\right)^2$$

$$= \frac{1}{1 + 2B_\beta\sqrt{\theta} + B_\beta^2\theta} \geq 1 - 2B_\beta\sqrt{\theta} - B_\beta^2\theta,$$

Overall, we proved the lemma.

## 1.6 Proof of Lemma 2.2

By definition,
$$B = \sup_{\|\mathbf{v}\|_2 \leq 1} \mathbb{E}\left[|\langle \mathbf{v}, X\rangle|^4\right] \geq \mathbb{E}\left[\left|X^j\right|^4\right], \ \forall j = 1, 2, \cdots, d,$$

where $X^j$ denotes the $j$-th entry of the random vector $X$. Also, for any fixed vector $\mathbf{v} \in \mathbb{R}^d$, we have

$$0 \leq \mathbb{E}\left[\left(|\langle \mathbf{v}, X\rangle|^2 - \left|X^j\right|^2\right)^2\right] = \mathbb{E}\left[|\langle \mathbf{v}, X\rangle|^4\right] + \mathbb{E}\left[\left|X^j\right|^2\right] - 2\mathbb{E}\left[|\langle \mathbf{v}, X\rangle|^2\left|X^j\right|^2\right]$$

$$\Rightarrow \mathbb{E}\left[|\langle \mathbf{v}, X\rangle|^4\right] + \mathbb{E}\left[\left|X^j\right|^2\right] \geq 2\mathbb{E}\left[|\langle \mathbf{v}, X\rangle|^2\left|X^j\right|^2\right], \ \forall j = 1, 2, \cdots, d.$$

Taking the supremum from both sides of the above inequality and use the previous bound on $B$, we get

$$\sup_{\|\mathbf{v}\|_2 \leq 1} \mathbb{E}\left[|\langle \mathbf{v}, X\rangle|^4\right] \geq \sup_{\|\mathbf{v}\|_2 \leq 1} \mathbb{E}\left[|\langle \mathbf{v}, X\rangle|^2\left|X^j\right|^2\right], \ \forall j = 1, 2, \cdots, d.$$

Summing over $i = 1, 2, \cdots, d$ gives

$$Bd = \sup_{\|\mathbf{v}\|_2 \leq 1} \mathbb{E}\big[|\langle \mathbf{v}, X\rangle|^4\big]d \geq \sum_{j=1}^{d} \sup_{\|\mathbf{v}\|_2 \leq 1} \mathbb{E}\Big[|\langle \mathbf{v}, X\rangle|^2 \big|X^j\big|^2\Big] \geq \sup_{\|\mathbf{v}\|_2 \leq 1} \mathbb{E}\Big[|\langle \mathbf{v}, X\rangle|^2 \|X\|^2\Big]$$

$$= \big\|XX^T\|X\|_2^2\big\| = \sigma_0^2.$$

## 1.7  Proof of Lemma 2.3

First of all, let $Z = X - \mu_0$, then, we have $\mathbb{E}[Z] = 0$. The lower bound of $\sigma_0^2$ follows directly from Corollary 5.1. It remains to show the upper bound. Note that by Cauchy-Schwarz inequality,

$$\sigma_0^2 = \big\|ZZ^T\|Z\|_2^2\big\| = \sup_{\|\mathbf{v}\|_2 \leq 1} \mathbb{E}\big[\langle Z, \mathbf{v}\rangle^2 \|Z\|_2^2\big]$$

$$\leq \sup_{\|\mathbf{v}\|_2 \leq 1} \mathbb{E}\big[\langle Z, \mathbf{v}\rangle^4\big]^{1/2} \mathbb{E}\big[\|Z\|_2^4\big]^{1/2}.$$

We then bound the two terms separately. For any vector $\mathbf{x} \in \mathbb{R}^d$, let $x^j$ be the $j$-th entry. Note that

$$\mathbb{E}\big[\langle Z, \mathbf{v}\rangle^4\big]^{1/2} = \left(\sum_{j=1}^{d} \mathbb{E}\big[(Z^j v^j)^4\big] + \sum_{j,k,\ j\neq k}^{d} \mathbb{E}\big[(Z^j v^j)^2 (Z^k v^k)^2\big]\right)^{1/2}$$

$$\leq \left(\sum_{j=1}^{d} \mathbb{E}\big[(Z^j v^j)^4\big] + \sum_{j,k,\ j\neq k}^{d} \mathbb{E}\big[(Z^j v^j)^4\big]^{1/2} \mathbb{E}\big[(Z^k v^k)^4\big]^{1/2}\right)^{1/2}$$

$$= \sum_{j=1}^{d} \sqrt{\mathbb{E}[(Z^j v^j)^4]} = \sum_{j=1}^{d} \sqrt{\mathbb{E}[(Z^j)^4]}(v^j)^2 \leq R \sum_{j=1}^{d} \mathbb{E}\big[(Z^j)^2\big](v^j)^2$$

where the last inequality uses the fact that the kurtosis is bounded. Thus,

$$\sup_{\|\mathbf{v}\|_2 \leq 1} \mathbb{E}\big[\langle Z, \mathbf{v}\rangle^4\big]^{1/2} \leq R \cdot \sup_{\|\mathbf{u}\|_1 \leq 1} \sum_{j=1}^{d} \mathbb{E}\big[(Z^j)^2\big]u^j = R \cdot \max_{j=1,2,\cdots,d} \mathbb{E}\big[(Z^j)^2\big] \leq R\|\Sigma_0\| \quad (10)$$

where the last inequality follows from

$$\max_{j=1,2,\cdots,d} \mathbb{E}\big[(Z^j)^2\big] = \max_{j=1,2,\cdots,d} \mathbf{e}_j^T \Sigma_0 \mathbf{e}_j \leq \sup_{\|\mathbf{v}\| \leq 1} \mathbf{v}^T \Sigma_0 \mathbf{v} = \|\Sigma_0\|.$$

Similarly, we have

$$\mathbb{E}\big[\|X\|_2^4\big]^{1/2} = \left(\sum_{j=1}^{d} \mathbb{E}\big[(X^j)^4\big] + \sum_{j,k=1,\ j\neq k}^{d} \mathbb{E}\big[(X^j)^2 (X^k)^2\big]\right)^{1/2}$$

$$\leq \left(\sum_{j=1}^{d} \mathbb{E}\big[(X^j)^4\big] + \sum_{j,k=1,\ j\neq k}^{d} \mathbb{E}\big[(X^j)^4\big]^{1/2} \mathbb{E}\big[(X^k)^4\big]^{1/2}\right)^{1/2}$$

$$\leq \sum_{j=1}^{d} \sqrt{\mathbb{E}[(X^j)^4]} \leq R \cdot \sum_{j=1}^{d} \mathbb{E}\big[(X^j)^2\big] = R \cdot \mathrm{tr}(\Sigma_0)$$

Combining the above bounds with (23) gives

$$\sigma_0^2 \leq R^2 \|\Sigma_0\| \mathrm{tr}(\Sigma_0),$$

which readily implies the lemma.