[Reviews · NeurIPS 2017]

Reviewer 1



The paper considers covariance matrix estimation in high-dimensions under heavy-tailed and non-centered data. Since the data is non-centered, the mean has to be estimated (under heavy-tail assumptions) to certain required accuracy and plugged-in to estimate the covariance matrix. Furthermore, the estimation method requires knowledge of certain parameters of the true covariance matrices. In order to overcome that, the paper proposes an adaptive procedure based on Lepski's method to automatically select the required parameters. The results are interesting and enables practical covariance matrix estimation high-dimensions. The main drawback of the paper is that most of the techniques used are already proposed in the literature and well-known (at least to researchers working in this area). Hence essentially this paper combines such techniques (in a slightly non-trivial way) to obtain the required results.

Reviewer 2



The paper provides a truncation estimator for the covariance matrix. The idea is simple yet useful, enjoying very good theoretical properties. I strongly welcome this contribution to the robust statistics literature. I have only several very minor comments/suggestions. (1) The authors should discuss relevance to covariance estimation under elliptical distributions, like ECA proposed by Han and Liu (JASA, 2017+) (which is also a robust estimator of the covariance when the eigenvalues are bounded; see, e.g., https://arxiv.org/pdf/1705.06427.pdf). See below for more details. (2) Can we derive estimation accuracy under other norms (such as L_infty norm and restricted spectral norm, defined in the above ECA paper)? These alternative norms are crucial in high dimensional statistics (e.g., graphical model estimation and sparse PCA). See the ECA paper for an example. (3) Centeredness is not crucial in covariance matrix estimation as long as the data are iid: We could use the pairwise difference X_i-X_j, which is centered no matter whether X_i is centered or not. This alternative should also be more efficient than the proposed one since it avoids estimating the mean. See again the ECA paper for discussions. Relative comments have also been put in Chen, Gao, and Ren (AoS, 2017+), which the authors cited.

Reviewer 3



This paper studies robust estimation of covariance matrix using adaptive truncation, where existing results work for centered data. The main contribution is to control the additional error incurred by the use of a plug-in mean estimate (Lemma 4.2), which requires a bit technical work. The results a valid and make a fine contribution to this topic. This paper provides results on the error bounds in terms of the intrinsic dimension, which is useful and interesting. However, the claim in the abstract that the "results are applicable in the case of high-dimensional observations" is a bit far-fetching, as the results are useful only when the intrinsic dimension is low, which is not the case if the data is from a sparse high dimensional model, such as in the case of sparse PCA (intrinsic dimension is high, but only a few relevant variables). Typo: the math display right above line 220, there is a missing supremum over $\mathbf{v}$ .